

# Fast and accurate face recognition system using MORSCMs-LBP on embedded circuits

Khalid M. Hosny[1], Aya Y. Hamad[2], Osama Elkomy[1] and Ehab R. Mohamed[1]

[1] Department of Information Technology, Faculty of Computers and Informatics, Zagazig University, Zagazig, Egypt
[2] Department of Information Technology/Information Technology and Computer Science, Sinai University, North Sinai, Al Arish, Egypt

## ABSTRACT

Because of the current COVID-19 circumstances in the world and the tremendous technological developments, it has become necessary to use this technology to combat the spread of the new coronavirus. The systems that depend on using hands, such as fingerprint systems and PINs in ATM systems, could lead to infection, so they have become undesirable and we can replace them by using facial recognition instead. With the development of technology and the availability of nano devices like the Raspberry Pi, such applications can be implemented easily. This study presents an efficient face recognition system in which the face image is taken by a standalone camera and then passed to the Raspberry Pi to extract the face features and then compare them with the database. This approach is named MORSCMs-LBP by combining two algorithms for feature extraction: Local Binary Pattern (LBP) as a local feature descriptor and radial substituted Chebyshev moments (MORSCMs) as a global feature descriptor. The significant advantage of this method is that it combines the local and global features into a single feature vector from the detected faces. The proposed approach MORSCMs-LBP has been implemented on the Raspberry Pi 4 computer model B with 1 GB of RAM using C++ OpenCV. We assessed our method on various benchmark datasets: face95 with an accuracy of 99.0278%, face96 with an accuracy of 99.4375%, and grimace with 100% accuracy. We evaluated the proposed MORSCMs-LBP technique against other recently published approaches; the comparison shows a significant improvement in favour of the proposed approach.

# INTRODUCTION

Since the beginning of creation, people have been recognized by their faces, and over time, there has been a significant breakthrough in the realm of technology, resulting in the construction of a smart society. Face recognition is considered the most challenging pattern recognition application because some conditions can affect the face image, such as illumination, scaling, occlusion, and other factors. It has attracted significant attention because it can identify and verify a person in a digital photo or video frame. Face recognition has been used in many fields, such as helping police find suspects and missing

Corresponding author
Aya Y. Hamad,
ayaraslan46@gmail.com

persons, forensic medicine to identify the identity through the scanned image, and for security purposes. Security cameras can now be used in ATMs, banks, universities, homes, airports, offices, and many other fields (*Suresh Madhavan, 2019*).

All the previously mentioned usages and the availability of nano devices such as the Raspberry Pi have motivated researchers to introduce more accurate and fast approaches to achieve it. *Mustakim et al. (2019)* presented a face recognition system that acquires the image from the video stream. Then the database is searched for any matches with the input image. If the person was found, his/her name and additional information are displayed.

*Awais et al. (2019)* also produced Face Recognition for Real-Time Surveillance. The images were taken from the video stream and then they were compared with the images stored in the database. If there is no match with the database, an alarm or signal is sent out by the security system to take the proper action.

*Lee et al. (2020)* presented face recognition for access control using a standalone camera. They used the (LBP)-AdaBoost algorithm for face and eye detection, and then the face features were extracted by replacing the Gabor-LBP histogram with Gaussian. Because all previous works convert the face image to grayscale, some data is lost, which may affect the recognition accuracy.

*Ben et al. (2021)* discussed different approaches to differentiate between macro and micro-expressions, they also presented a new dataset called MMEW, that has more videos and labelled emotion classes.

The Raspberry Pi was employed in various face recognition applications because of its superiority and portability (*Gunawan et al., 2017*), a door unlocking system using Raspberry Pi (*Vamsi, Sai & Vijayalakshmi, 2019*; *Selvaraj, Alagarsamy & Dhilipkumar, 2021*), *Sajjad et al. (2020)* suggested a system that helps the police to find missing persons and suspects using wireless cameras, *Hasban et al. (2019)* proposed an attendance system for the classroom, *Ambre, Masurekar & Gaikwad (2020)* produced a face recognition system using Raspberry Pi that can be used by governments to save money for CCTV monitoring staff (*Gaikwad, 2020*), *Majid et al. (2021)* proposed a PCA method for face recognition, but the best recognition rate they achieved was 82.5%.

Face recognition has many ways to be implemented, but they compare extracted facial features from a specific image with the facial extracted features stored in a database. This technology is used extensively for security purposes but is also used in many other fields. Whatever the application is, it is supposed to be accurate and fast. In this study, we developed a speedy, accurate, and simple method for face detection and recognition. The method depends on the global and local features of the image, which provide sufficient information to recognize the face. First, the face recognition system comprises two stages. The first is face detection, in which the image is searched for the face region, and then this region is cropped to make the recognition process easier.

There are various types of algorithms for face detection in a digital image or video frame. In this approach, we used the Viola-Jones algorithm, the most common face detection introduced by *Viola (2001)*. It has been widely used in a variety of works,

including masked-face detection (*SivaKumar et al., 2021*), detecting drowsy drivers (*Rajendran et al., 2021*; *Fatima et al., 2020*), and many other applications.

Orthogonal moment (OMs) for color images is a widespread topic. A novel color face recognition method called multi-channel orthogonal fractional-order moment proposed by *Hosny, Abd Elaziz & Darwish (2021)*, and this method proved its efficiency, invariance to transformation, and robustness to noise. Orthogonal moments proved its efficiency in various pattern recognition applications such as color image watermark (*Darwish, Hosny & Kamal, 2020*), *Ranade & Anand (2021)* proposed color face recognition technique based on Zernike quaternion moment vector and using quaternion vector moment (QVM) similarity distance to enhance accuracy, and the experiment proved its superiority on all other techniques. Abdelmajid et al. introduced a face recognition system based on quaternion moment and deep neural network (DNN), it is computationally low cost and also accurate (*Alami et al., 2019*). *Hosny (2019)* proposed another method for face recognition using exact Gaussian-hermit moments and it could overcome geometric distortions.

The rest of this paper is arranged as follows: materials and methods, then experimental results and discussion and finally conclusion.

## MATERIALS AND METHODS

### Chebyshev rational moments for gray level images

The Chebyshev OMs computations were illustrated by *Guo et al. (2017)* as:

MORSCMs was derived by *Guo et al. (2017)*, For a color image, the multi-channel moments are a set OMs of each color channel, so the =MORSCMs= for the (R, G, and B) channels are as:

$$x_{mn} = \frac{1}{2\pi\alpha_m} \int_0^{2\pi} f_j(r,\theta) R_m(r) w_i(r) e^{-\hat{i}n\theta} r\, dr\, d\theta \tag{1}$$

where m, n are non-negative integers for order and repetition, $\hat{i} = \sqrt{-1}$, the equation of weight function, Wi (r) is:

$$Wi(r) = \frac{1}{(1+r)\sqrt{r}} \tag{2}$$

The Rm (r) is the Chebyshev rational polynomials defined as:

$$Rm(r) = \sum_{k=0}^m \alpha_{mk} \frac{1}{(1+r)^k} \tag{3}$$

where $\alpha_{mk}$ defined as:

$$\alpha_{mK} = \frac{(k!)^2}{(2k)!} \binom{m+k-1}{k} \binom{m}{k} (-4)^\kappa \tag{4}$$

The orthogonality condition of Chebyshev rational moments can be expressed by multiplying polynomials $R_m(r)$, $R_n(r)$ and $W(r)$ over the interval $[0,\infty)$ as:

$$\int_0^\infty R_m(r)\, R_n(r)\, W(r)\, dr = \delta_{mn}\frac{A\pi}{2} \tag{5}$$

where, $\delta_{mn} = \begin{cases} 1, & \text{for } m = n \\ 0, & \text{otherwise} \end{cases} \tag{6}$

and $\frac{x_m\pi}{2}$ is the normalization constant.

$$A_m = \begin{cases} 1, & \text{if } m = 0 \\ 2, & \text{if } m \geq 1 \end{cases} \tag{7}$$

B. Multi-channel radial substituted Chebyshev moments.
This method was introduced by *Hosny (2019)*, where Rm (r) is replaced with R$\bar{m}$ ($\hat{r}$)

$$\hat{r} = \frac{\hat{r}}{1+\hat{r}} \text{ and } dr \rightarrow \frac{d\hat{r}}{(1-\hat{r})^2} \tag{8}$$

and radial substituted Chebyshev moment function R$\bar{m}$ ($\hat{r}$) defined as:

$$\text{R}\bar{\text{m}}(\hat{r}) = \sum_{k=0}^m \frac{(k!)^2}{(2k)!}\binom{m+k-1}{k}\binom{m}{k}(-4)^k\frac{1}{(1+\hat{r})^k} \tag{9}$$

for $m \geq 1$
And by using recurrence relation R$\bar{m}$($\hat{r}$) is defined as:

$$\text{R}\bar{\text{m}}+1(\hat{r}) = 2\,(2\hat{r}-1)\,\text{R}\bar{\text{m}}(\hat{r}) - \text{R}\bar{\text{m}}-1(\hat{r}) \quad \text{for } m \geq 1 \tag{10}$$

And $\bar{R}0\,(\hat{r}) = 1$, $\bar{R}1\,(\hat{r}) = 2\hat{r}-1$
Then substituted weight function Wi ($\hat{r}$)

$$\text{Wi}(\hat{r}) = \frac{1}{(1=\hat{r})\sqrt{\hat{r}}} = \frac{(1-\hat{r})^{1.5}}{(\sqrt{\hat{r}}+\hat{r})^2} = \frac{1}{(\hat{r}-\hat{r}^2)^{0.5}} \tag{11}$$

So, the R$\bar{m}$ ($\hat{r}$) is orthogonal over unit disk on the interval [0, 1] as:

$$\int_0^1 \text{R}_m(\hat{r})\,\text{R}_n(r)\,\text{W}(\hat{r})\,d\hat{r} = \left(\frac{x_m\pi}{2}\right)\delta_{mn} \tag{12}$$

For a color image, the multi-channel moments are described as set OMs of each color channel (*Hosny & Darwish, 2019*), so the MORSCMs for the (R, G, and B) channel areas:

$$x_{mn}(f_r) = \frac{1}{2\pi\alpha_m}\int_0^{2\pi}\int_0^1 f_r(\hat{r},\theta)\text{R}_m(\hat{r})\text{W}_i(\hat{r})e^{-\hat{i}n\theta}\hat{r}\,d\hat{r}\,d\theta \tag{13}$$

$$x_{mn}(f_g) = \frac{1}{2\pi\alpha_m}\int_0^{2\pi}\int_0^1 f_g(\hat{r},\theta)\text{R}_m(\hat{r})\text{W}_i(\hat{r})e^{-\hat{i}n\theta}\hat{r}\,d\hat{r}\,d\theta \tag{14}$$

$$x_{mn}(f_b) = \frac{1}{2\pi\alpha_m} \int_0^{2\pi} \int_0^1 f_b(\hat{r},\theta) R_m(\hat{r}) W_i(\hat{r}) e^{-\hat{i}n\theta} \hat{r}\, d\hat{r}\, d\theta \qquad (15)$$

where the $f_r(\hat{r},\theta)$, $f_g(\hat{r},\theta)$, $f_b(\hat{r},\theta)$ respectively, show intensity values of the three red, green, and blue channels.

### 1) Rotation invariant

The MORSCMs method is invariance to rotation, which means if the image is rotated by an angle ($\alpha$), then:

$$f_{color}^{rot}(\hat{r},\theta) = f_{colur}(\hat{r},\theta+\alpha) \qquad (16)$$

consider $\hat{\theta} = (\theta+\alpha)$ so $\theta = (\hat{\theta}-\alpha)$ and $d\hat{\theta} = d\theta$

$$x_{mn}(f_{color}^{rot}) = \frac{1}{2\pi\alpha_m} \int_0^{2\pi} \int_0^1 f_c(\hat{r},\theta+\alpha) \overline{R}_m(\hat{r}) W_i(\hat{r}) e^{-\hat{i}n\theta} \hat{r}\, d\hat{r}\, d\theta$$

$$= \frac{1}{2\pi\alpha_m} \int_0^{2\pi} \int_0^1 f_c(\hat{r},\theta+\alpha) \overline{R}_m(\hat{r}) W_i(\hat{r}) e^{-\hat{i}n(\hat{\theta}-\alpha)} \hat{r}\, d\hat{r}\, d\hat{\theta}$$

$$= \frac{1}{2\pi\alpha_m} \int_0^{2\pi} \int_0^1 f_c(\hat{r},\theta+\alpha) \overline{R}_m(\hat{r}) W_i(\hat{r}) e^{-\hat{i}n\hat{\theta}} \; e^{\hat{i}n\alpha} \hat{r}\, d\hat{r}\, d\hat{\theta}$$

$$= x_{mn}(f_{color}^{rot}) e^{\hat{i}n\alpha} \qquad (17)$$

where color $\in$ {*red, green, blue*}

so, depending on Eq. (17):

$$|e^{\hat{i}n\alpha}| = 1$$

so, $|x_{m,n}(f_{color}^{rot})| = |x_{mn}(f_{color}^{rot})\, e^{\hat{i}n\alpha}|$

then,

$$|x_{m,n}(f_{color}^{rot})| = |x_{m,n}(f_{color}^{rot})| \qquad (18)$$

### 2) Scaling invariant

OMs values are invariant to scaling, as the calculation region can cover the whole contents of the image as explained in *Guo et al. (2017)*, so the original image is defined on the unit circle as shown in Fig. 1.

### 3) Translation invariant

$$A_c = (p_{10}(f_r) + p_{10}(f_g) + p_{10}(f_b))/p_{00}$$
$$B_c = (p_{01}(f_r) + p_{01}(f_g) + p_{01}(f_b))/p_{00} \qquad (19)$$
$$P_{00} = P_{00}(f_r) + P_{00}(f_g) + P_{00}(f_b)$$

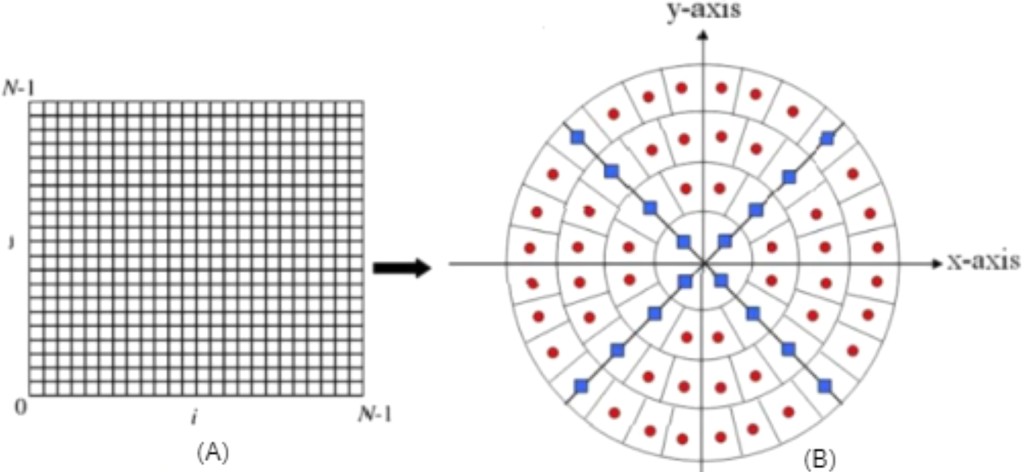

**Figure 1 (A) Original image (cartesian pixel), (B) converted image (polar pixel).**

where (Ac, Bc) illustrates the color image centre point and P01( ƒt), P00( ƒt), P10( ƒt) represent the zero and first order of orthogonal moment for each channel sequentially. So, the MORSCMs is invariance to translation and could be calculated as:

$$\bar{x}_{mn}(f_c) = \frac{1}{2\pi\alpha_m}\int_0^{2\pi}\int_0^1 f_c(\bar{r},\bar{\theta})\overline{R}_m(\bar{r})W_i(\bar{r})e^{-inr^-}\bar{r}d\bar{r}d\bar{\theta} \qquad (20)$$

where the $(\bar{r}, \bar{\theta})$ is the image pixel after moving it to the centroid (Ac, Bc).

### Local binary pattern

LBP considered texture descriptor. It has become a widespread approach because of its simplicity and discriminative power. It was first described for texture classification by *Ojala, Pietikainen & Maenpaa (2002)*. It can be computed using the following formula: It works by thresholding the neighbourhood of each pixel in the image, and then it converts them to binary numbers, as shown.

$$LBP(x, y) = \sum_{n=0}^{7} C(I_n - I_{thrsh})2^n \qquad (21)$$

where,

$$C(r) = \begin{cases} 1 & r \geq 0 \\ 0 & otherwise \end{cases}$$

and $I_{thrsh} = I(x, y)$, is the central pixel.
Then the LBPH is used and calculated as:

$$LBPH(t) = \sum \delta\{t, LBP(x, y)\}, \quad t = \{0, ...., 7\} \qquad (22)$$

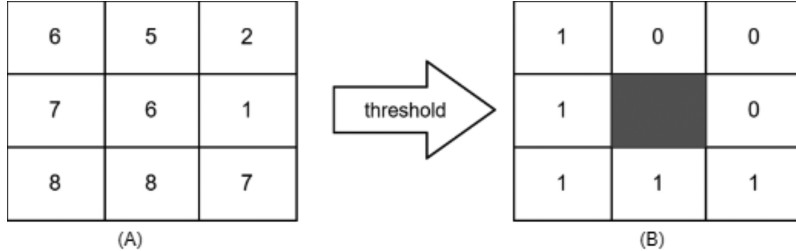

**Figure 2** (A) 3 × 3 neighborhood image window, (B) threshold results binary code = 11110001 and LBP = 1 + 16 + 32 + 64 + 128 = 241.    

But it is improved by *Hosny & Darwish (2019)* to use neighbors of different sizes to take any radius and neighbors that surround the central pixel. It is called a uniform local binary pattern, and it presents less feature vector length. It is computed as:

$$\text{LBP}_{n,r}(I_{thrsh}) = \begin{cases} \sum_{n=0}^{n-1} C(I_n - I_{thrsh})2^n & if\ U(LBP_{n,r}) > 2 \\ n+1 & otherwise \end{cases} \tag{23}$$

$$\text{LBPH}_n = \sum f\{I(x,y) = n\}, \quad n = \{0, ...., n-1\} \tag{24}$$

where r is the radius, n is the number of neighbours.

Figure 2 shows a sample of how LBP works.

## System architecture

The proposed method presents an efficient face detection and recognition system. The computational power is reduced and storage is done with no effect on the efficiency and accuracy. In this work, the face recognition system comprises four basic steps, arranged as:

1. Face detection for the input image using Viola-Jones, the most widely used operator.
2. Cropping the face to facilitate feature extraction.
3. MORCMs and LBP are used to extract features.
4. Finally, the classification step on the extracted features is done using three popular standard classifiers. Figure 3 shows how the proposed algorithm works.

## Viola-Jones algorithm

Viola-Jones was first proposed in 2001 by *Viola (2001)*. There are many algorithms for face detection; detailed illustrations of these discussed approaches are available (*Fasel & Movellan, 2002*; *Srivastava et al., 2017*; *Seredin, 2010*). As proved by *Dang & Sharma (2017)*, Viola-Jones is the most widely used and efficient approach for detecting faces. The Viola-Jones algorithm can detect faces in various conditions like illumination, scaling, expression, pose, and makeup. It has been used in various works (*SivaKumar et al., 2021*; *Fatima et al., 2020*; *Ismael & Irina, 2020*; *Kirana, Wibawanto & Herwanto, 2018*; *Kumar, 2021*). After locating the region of the face, it is cropped to minimize the region of interest. Then, the features of the face area are extracted using MORCMs and LBP. The three popular standard classifiers (KNN, SVM, and simple logistic) are employed to classify the extracted features (see Fig. 4).

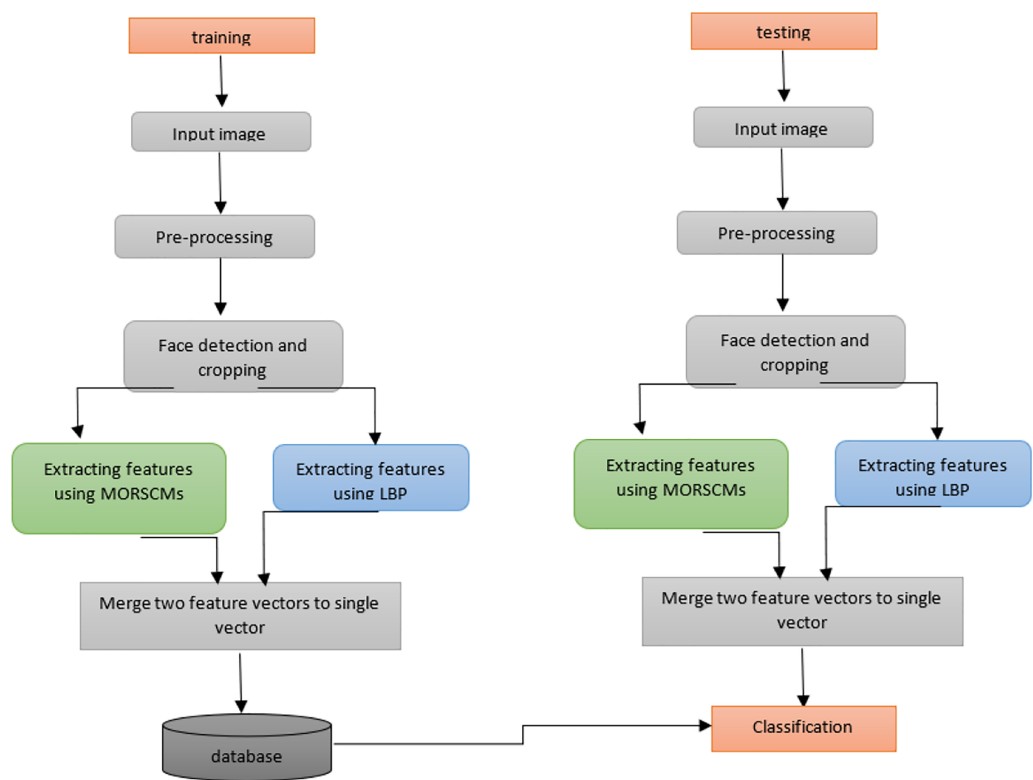

**Figure 3** Diagram shows the whole system for color face recognition and classification.

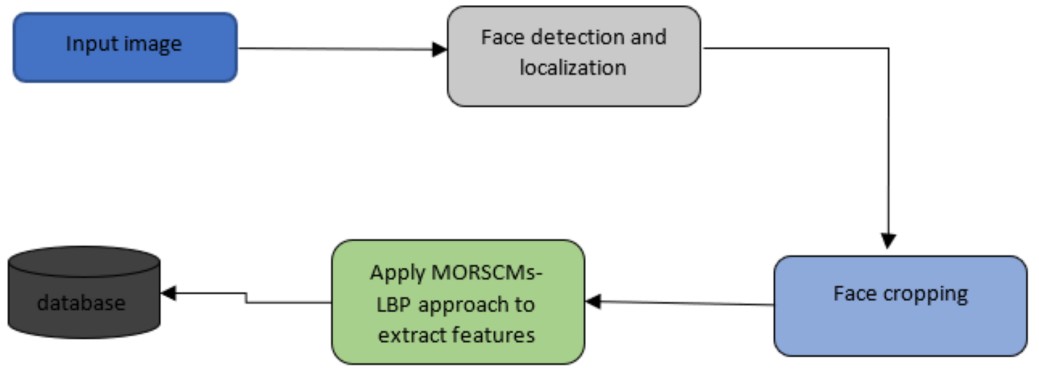

**Figure 4** Schematic representation for the input image to detect and crop the face.

## Feature extraction using MORSCMs-LBP

MORSCMs method was introduced by *Hosny & Darwish (2019)* as a global feature descriptor for color image representation and recognition. It has proven to be effective in terms of recognition accuracy. In the proposed approach MORSCMs-LBP, we combined both methods: MORSCMs and LBP. LBP was used as a local texture features descriptor for the image. The global shape features are extracted using MORCMs. The MORSCMs method was introduced by *Hosny & Darwish (2019)* as a global feature

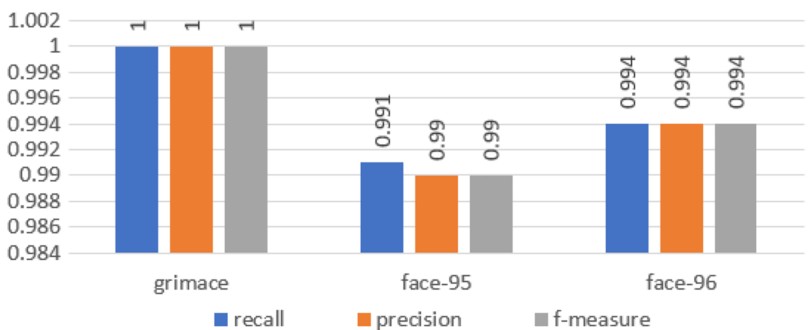

**Figure 5  Chart of the performance metrics for face95, face96 and grimace datasets.**

descriptor for color image representation and recognition. It has proven to be effective in terms of recognition accuracy. In the proposed approach (MORSCMs-LBP), we combined both methods: MORSCMs and LBP. LBP was used as a local texture feature descriptor for the image. The global shape features are extracted using MORCMs. The two feature vectors are merged to get one feature vector, which is used for classification. See Fig. 3.

The MORSCMs-LBP implemented over three distinct challenging datasets (faces95, face96, grimace). A detailed explanation of these datasets is introduced in the next section. The most significant advantage of the proposed approach is that it requires less storage capacity and less computational cost, suitable for modern systems such as raspberry pi, compared to other methods.

## EXPERIMENTAL RESULTS AND DISCUSSION

As mentioned earlier, this study aims to provide a color face recognition system that works efficiently and takes less computational time. We performed three experiments to evaluate the proposed method. These experiments were performed using three distinct standard datasets (faces 95, faces96, and grimace). We used different moment orders (5, 10, and 15) to test accuracy and reliability. The accuracy is measured using the following equation:

$$\mathbf{Accur} = (\mathbf{T_p} + \mathbf{T_n})/(\mathbf{T_p} + \mathbf{T_n} + \mathbf{F_p} + \mathbf{F_n}) \tag{25}$$

where $Tp$, $Tn$, $Fp$ and $Fn$ are the number of true-positive classified, true-negative classified, false-positive classified, and false-negative classified, respectively. We also computed other efficiency measures such as precision, F-measure, recall, specificity, and they are defined as:

$$Precision = (Tp)/(Tp + Fp) \tag{26}$$
$$Recall = (Tp)/(Tp + Fn) \tag{27}$$
$$F - measure = 2 * Precision * Recall/(Precision + Recall) \tag{28}$$

We calculated all these metrics for our approach and the results are illustrated in Fig. 5.

**Table 1 Raspberry pi specification for face recognition.**

| Name | Configuration |
|---|---|
| Operating system | AKA Raspbian |
| Programming language | C++ |
| Libraries | Opencv 4.1.0, Cmath, Dlib, Complex |

**Table 2 Differences between face 95, face 96, and grimace datasets.**

| Dataset | Faces 95 | Faces 96 | Grimace |
|---|---|---|---|
| Resolution | 180 × 200 | 196 × 196 | 180 × 200 |
| No. classes | 72 | 152 | 18 |
| Total images | 1,440 | 3,040 | 360 |
| Background | Red curtain | Complex (glossy posters) | Plain background |
| Head scale | Large variation | Large variation | Small variation |
| Lighting variation | Significant lighting change | Significant lighting change | Very little change |
| Translation | Large variation | Some variation | Some variation |
| Expression variation | Some variation | Some variation | Major variation |

## System configuration

The experiments were executed on a Raspberry Pi, which is used to acquire the input face image. The used Raspberry Pi contains a Broadcom BCM2711 system on a chip with a 1.5 GHz processor and an ARM Cortex-A72. The GPU of the Raspberry Pi is the VI GPU. OpenGL ES2.0 and OpenCV libraries are used to access fast 3D cores. Several ports are present on the Raspberry Pi 4 Model B board, including two micro-HDMI ports, two USB 3.0 ports, two USB 2.0 ports, and a Gigabit Ethernet port. A total of 1 GB RAM on the B1 computer the default operating system for the Raspberry Pi (aka Raspbian) is built on top of Debian. We also tested a dual Intel (R) Corei7 7700 QH (2.80 GHz) CPU. The used operating system is Windows 10, and the IDE is Visual Studio 2019. Table 1 shows more details about the configuration and hardware specifications of the used device.

## Face-95, faces-96, and grimace dataset

We used three different benchmark datasets to perform our experiments and will show some sample images of them. A sequence of 20 images was taken for each person using a fixed camera and 0.5 s between each taken image. During taking the sequence, the person moves one step toward the camera to get different scales. The images in each dataset vary in background, head scale, translation, facial emotions, and image lighting. See Table 2. The used dataset was introduced by Spacek (2009).

## Experiment evaluation

In the proposed method, we used MORSCMs-LBP to extract facial features. We used the LBP with 59 features and different moment orders of five, 10, and 15. The number of extracted features is 31, 111, and 241, respectively. The LBP features vector was merged

**Table 3 The accuracy of recognition for the proposed method compared with other state of art methods using standard grimace dataset.**

| Acc | QALMs (*Kumar, 2021*) | QCMs (*Spacek, 2009*) | QKrMs (*Kirana, Wibawanto & Herwanto, 2018*) | QMV (*Hassan et al., 2021*) | QPZMV (*Hassan et al., 2021*) | Proposed |
|---|---|---|---|---|---|---|
| | 96.2963% | 94.0741% | 97.0370% | 96.67% | 94.44% | 100% |
| **Gain** | 3.7037 | 5.9259 | 2.963 | 3.33 | 5.56 | 0 |

Note:
gain = proposed approach – compared method.

**Table 4 The accuracy of recognition for the proposed method compared with other state of art methods using standard face95 dataset.**

| Acc | FCNN (*Vinay et al., 2017*) | GCNN (*Vinay et al., 2017*) | HOG (*Girdhar, Virmani & Kumar, 2019*) | FrMEMs (*Hosny, Abd Elaziz & Darwish, 2021*) | Openface (*Ayoub et al., 2021*) | Inspection-V3 (*Ayoub et al., 2021*) | Proposed |
|---|---|---|---|---|---|---|---|
| | 92.21% | 92.44% | 91.5% | 92 % | 99% | 95% | 99.0278% |
| **Gain** | 6.8178 | 6.5878 | 7.5278 | 7.0278 | 0.0278 | 4.0278 | 0 |

Note:
gain = proposed approach – compared method.

**Table 5 The accuracy of recognition for the proposed method compared with other state of art methods using standard face96 dataset.**

| Acc | HOG (*Girdhar, Virmani & Kumar, 2019*) | FrMEMs (*Hosny, Abd Elaziz & Darwish, 2021*) | QKrMs (*Virmani et al., 2019*) | Openface (*Ayoub et al., 2021*) | Inspection-V3 (*Ayoub et al., 2021*) | Proposed |
|---|---|---|---|---|---|---|
| | 91.5% | 99% | 99.39% | 97% | 97% | 99.4375% |
| **Gain** | 7.9375 | 0.4375 | 0.0475 | 2.4375 | 2.4375 | 0 |

Note:
gain = proposed approach – compared method.

with MORSCMS's features vector to produce a single feature vector as shown in Fig. 3, so the total number of features extracted was 90, 170, and 300, respectively for each moment order. The proposed face recognition system is tested over a very challenging dataset, as illustrated in the earlier section. The accuracy of the obtained results has been confirmed by popular and standard classifiers; the k-nearest neighbor with k = 1, support vector machine, and simple logistic.

The experiments proved that our approach is more efficient than other currently proposed approaches. *Hassan et al. (2021)* proposed a descriptor for face recognition called (QKrMs); *Pan, Li & Zhu (2015)* proposed (QALMs); *Pan, Li & Zhu (2020)* proposed another descriptor called (QCMs); and Sukhjeet et al. proposed (QMV) approach for color face recognition (*Ranade & Anand, 2021*); *Vinay et al. (2017)* presented two methods for facial recognition based on convolution neural network called (FCNN & GCNN); *Girdhar, Virmani & Kumar (2019)* proposed a hybrid Fuzzy based Behavior Prognostic System for Disparate traits (FBPSDs) which uses face recognition to identify the contextual faces; *Hosny, Abd Elaziz & Darwish (2021)* proposed a novel color face recognition method that depends on a new family of fractional-order orthogonal functions called (FrMEMs); *Virmani et al. (2019)* presented face detection and recognition approach based on deep learning called (FDREnet); *Ayoub et al. (2021)* presented Deep learning models "Openface *via* PSO and introduced customized Inception-V3 model *via* PSO.

The recognition rates of the proposed approach (MORSCMs-LBP) against the compared methods are depicted in Tables 3, 4, 5. The obtained results showed that MORSCMs-LBP was superior over the compared approaches.

**Table 6 A comparison between the number of features for the various values of moment order for the moment based approaches.**

| Moment-based approaches | Moment orders | | |
|---|---|---|---|
| | 5 | 10 | 15 |
| QZM | 12 | 36 | 72 |
| MZM | 37 | 108 | 217 |
| QZMV | 46 | 138 | 281 |
| QPZM | 21 | 66 | 136 |
| MPZM | 64 | 198 | 409 |
| QPZMV | 80 | 253 | 530 |
| MORSMs-LBP | 90 | 170 | 300 |

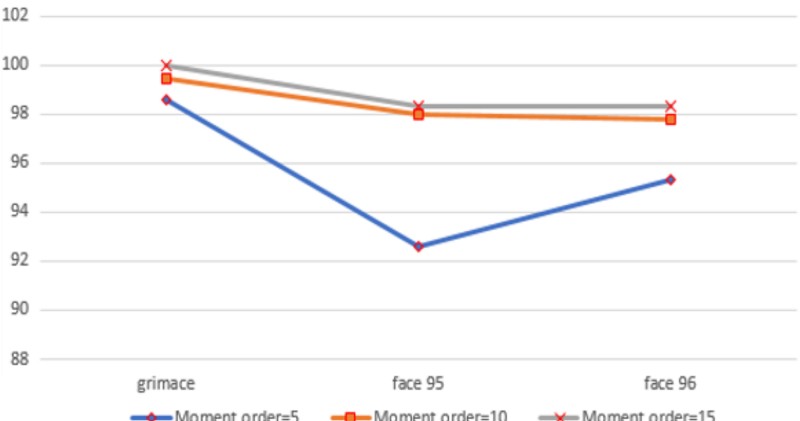

**Figure 6 The chart shows the recognition accuracy results for three datasets using moment orders five, 10 and 15.**

And Table 6 shows a comparison between the number of features for different moment-based approaches.

## RESULTS

In the first experiment, we used moment order five. Then we increased the moment to 10 to increase the number of extracted features, and finally, we set the moment value to 15. For moment order 15, we could get high accuracy and the number of extracted features is still small. We performed the three experiments on three benchmarked datasets and calculated the recognition rates using different standard classifiers. The results included the following parameters: Recall, precision and f-measure, which were illustrated through Eqs. (25)–(28), are shown in Fig. 5. A comparison between the accuracy results for three datasets using moments is illustrated in Fig. 6.

Figure 7 shows the confusion matrix for the KNN classifier with the grimace dataset using moment order 15. The confusion matrix shows that the KNN classifier could recognize all 20 images for each class (*i.e.*, 18 people).

| | And | Aant | Chr | Dah | Dav | Den | Glen | Ian | Jer | John | Lib | Mike | Pat | Sar | Ste | Stu | Tom | Will |
|---|---|---|---|---|---|---|---|---|---|---|---|---|---|---|---|---|---|---|
| And | 20 | | | | | | | | | | | | | | | | | |
| Ant | | 20 | | | | | | | | | | | | | | | | |
| Chr | | | 20 | | | | | | | | | | | | | | | |
| Dah | | | | 20 | | | | | | | | | | | | | | |
| Dav | | | | | 20 | | | | | | | | | | | | | |
| Den | | | | | | 20 | | | | | | | | | | | | |
| Glen | | | | | | | 20 | | | | | | | | | | | |
| Ian | | | | | | | | 20 | | | | | | | | | | |
| Jer | | | | | | | | | 20 | | | | | | | | | |
| John | | | | | | | | | | 20 | | | | | | | | |
| Lib | | | | | | | | | | | 20 | | | | | | | |
| Mike | | | | | | | | | | | | 20 | | | | | | |
| Pat | | | | | | | | | | | | | 20 | | | | | |
| Sar | | | | | | | | | | | | | | 20 | | | | |
| Ste | | | | | | | | | | | | | | | 20 | | | |
| Stu | | | | | | | | | | | | | | | | 20 | | |
| Tom | | | | | | | | | | | | | | | | | 20 | |
| will | | | | | | | | | | | | | | | | | | 20 |

**Figure 7 Confusion matrix for moment order 15 using KNN classifier for grimace dataset.**

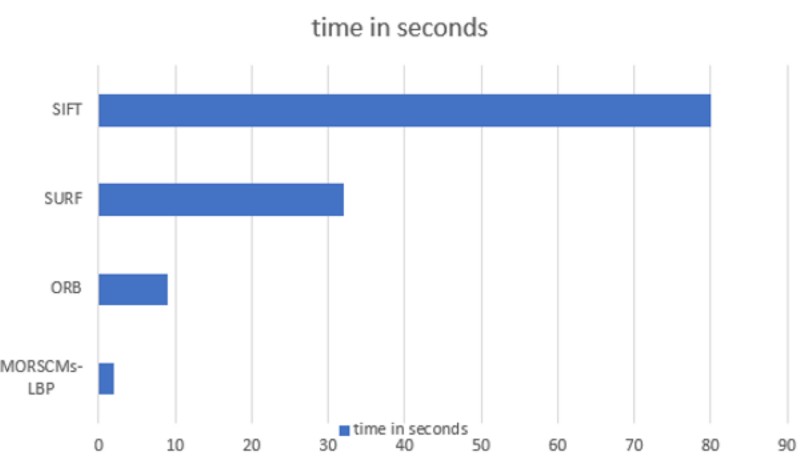

**Figure 8 Time analysis for SURF, ORB, SIFT and MORSCMs-LBP to recognize the image.**

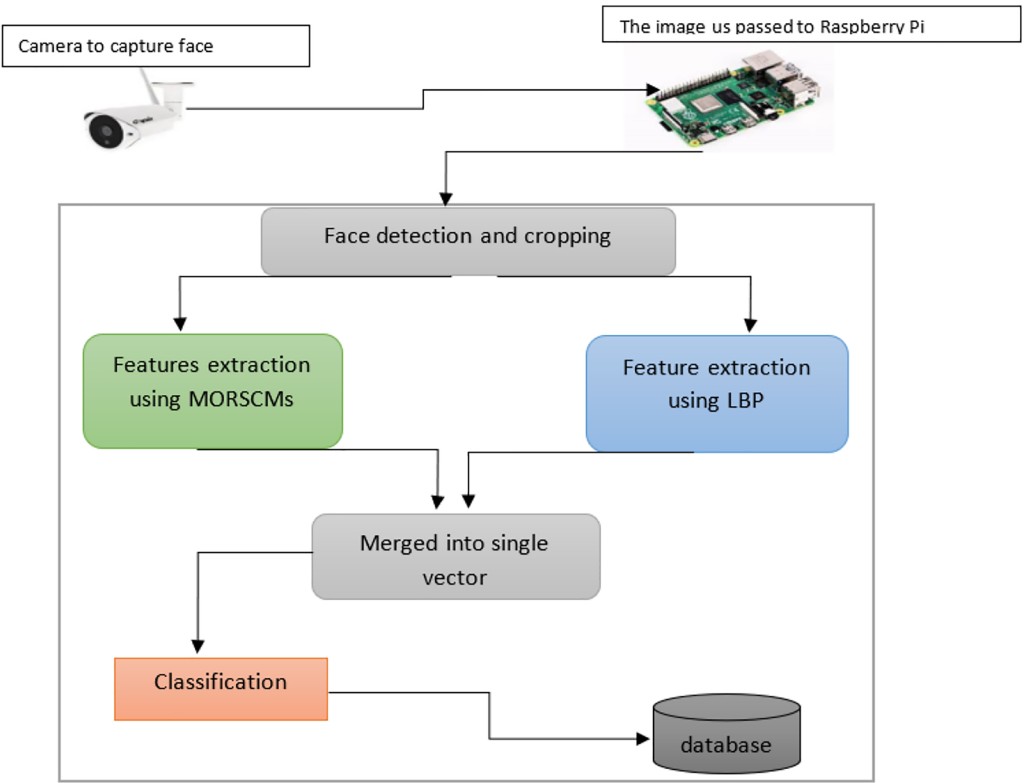

**Figure 9** **The diagram shows the whole system for color face recognition and classification.**

## Time analysis

Many factors affect the recognition time, such as the size of the image, the dataset's size, and the size of the feature vector. The length of the feature vector of the proposed approach was very small compared to the other compared approaches, where at moment order 15 (where the best recognition rate was obtained), it was 300, which is so small compared to the other approaches (*Sajjad et al., 2020*; *Karami, Prasad & Shehata, 2017*). It takes only 2 s to recognize the image, which is less than the time taken by the ORB, SIFT, and SURF approaches. Figure 8 shows the comparison between the recognition time of the proposed approach and the other compared methods.

## CONCLUSIONS

We proposed an efficient face recognition system based on MORSCMs-LBP where we integrated the global features of MORSCMs with the local LBP features to achieve a high recognition rate. MORSCMs are good shape feature descriptors where the basic functions of the MORSCMs can capture different unique types of information from the image according to the different values of moments order. This information may sometimes be an average intensity value, texture information, variance, and edge information in a different orientation. Also, many previous studies (*Karanwal & Diwaker, 2021*; *Guo et al., 2017*; *Wang et al., 2015*; *Suk & Flusser, 2009*; *Ojala, Pietikäinen & Harwood, 1996*; *Karami, Prasad & Shehata, 2017*; *Ahonen, Hadid & Pietikäinen, 2004*; *Hadid, 2008*;

*Huijsmans & Sebe, 2003; Grangier & Bengio, 2008; Ali, Georgsson & Hellstrom, 2008; Nanni & Lumini, 2008; Mäenpää, Viertola & Pietikäinen, 2003; Turtinen, 2006; Heikkila & Pietikainen, 2006; Kellokumpu, Zhao & Pietikäinen, 2008; Oliver et al., 2007; Kluckner et al., 2007; Fisher, Stein & Fisher, 2005; Sharma, Jain & Khushboo, 2019*) proved the high efficiency of the LBP as a local feature descriptor for the images. From the above results, we can observe that our proposed face recognition system has high accuracy and less recognition time, where the size of the extracted feature vector is relatively small compared with the other recently published methods (*Sajjad et al., 2020; Karami, Prasad & Shehata, 2017*). Also, we used the high efficiency of the LBP as a local feature descriptor for the face images. Besides the high accuracy, the low computational complexity of the proposed face recognition system makes it quick where the size of the extracted feature vector is relatively small compared with the other recently published methods. Figure 9 shows the schematic representation of the system. For more reliability, we applied the proposed algorithm using a nano device with the Raspberry Pi. The obtained results show high superiority in terms of accuracy and time when tested over three challenging datasets: Face-95, Face-96, and Grimace.

### Funding
The authors received no funding for this work.

### Competing Interests
The authors declare that they have no competing interests.

### Author Contributions
- Khalid M. Hosny conceived and designed the experiments, performed the computation work, authored or reviewed drafts of the article, and approved the final draft.
- Aya Y. Hamad conceived and designed the experiments, performed the experiments, analyzed the data, performed the computation work, prepared figures and/or tables, authored or reviewed drafts of the article, and approved the final draft.
- Osama Elkomy performed the computation work, prepared figures and/or tables, and approved the final draft.
- Ehab R. Mohamed analyzed the data, authored or reviewed drafts of the article, and approved the final draft.

### Data Availability
The images are available at Libor Spacek's Facial Images Databases: https://cmp.felk.cvut.cz/~spacelib/faces/.

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
