# Peer review of "Fast and accurate face recognition system using MORSCMs-LBP on embedded circuits"

_PeerJ Computer Science, doi:10.7717/peerj-cs.1008_

## Round 0.1 · original submission · Major Revisions

Some experiments should be added in this paper, therefore, this paper needs careful revisions.

Reviewer 1 ·

Basic reporting

Fails in 'Professional article structure, figures, tables. Raw data shared'.
All the figures and tables are not embedded in the manuscript.
So this manuscript is very hard to read and understand.

Experimental design

no comment

Validity of the findings

Fails in 'All underlying data have been provided; they are robust, statistically sound, & controlled'.
The proposed method was only compard with SOTA methods on the Grimace dataset.
Compared with other SOTA methods, how about the performance on the Faces 95 and Faces 96 datasets?

Additional comments

All the figures and tables are not embedded in the manuscript.
So this manuscript is very hard to read and understand.
The proposed method was only compard with SOTA methods on the Grimace dataset.
Compared with other SOTA methods, how about the performance on the Faces 95 and Faces 96 datasets?

Reviewer 2 ·

Basic reporting

The English language using is a little poor, and this paper t is very hard to read and understand. The authors may miss the related latest work, such as: Video-based Facial Micro-Expression Analysis: A Survey of Datasets, Features and Algorithms. IEEE Transactions on Pattern Analysis and Machine Intelligence, DOI: 10.1109/TPAMI.2021.3067464

Experimental design

None

Validity of the findings

The authors may miss the compared results on the Faces 95 and Faces 96 datasets.

Additional comments

None

---

## Round 0.2 · accepted · Accept

This paper has been well addressed the two reviewers' questions. This paper can be accepted now.

Reviewer 1 ·

Basic reporting

no comment

Experimental design

no comment

Validity of the findings

no comment

Additional comments

All of my concerns have been addressed. This version may be considered to be accepted.

Reviewer 2 ·

Basic reporting

no comment

Experimental design

no comment

Validity of the findings

no comment